# Provider-identified barriers to performance at seven Nigerian accident & emergency units: A cross-sectional study

**Muzzammil Imran Muhammad**[1]*, **Kelechi Umoga**[2], **Amber Acquaye**[1], **Brian Fleischer**[1], **Chigoziri Konkwo**[1], **Kehinde Olawale Ogunyemi**[3], **Christine Ngaruiya**[4]

**1** Yale School of Medicine, New Haven, Connecticut, United States of America, **2** Harvard Medical School, Massachusetts General Hospital/Brigham Women's Hospital, Cambridge, Massachusetts, United States of America, **3** Department of Community Medicine, Babcock University Teaching Hospital, Ilisham-Remo, Ogun State, Nigeria, **4** Department of Emergency Medicine, Yale University School of Medicine, New Haven, Connecticut, United States of America

* muzzammil.muhammad@yale.edu

**Data Availability Statement:** All data used for this work are available at: https://osf.io/ctz6h/.

## Abstract

### Background

Nigeria hosts much of Africa's morbidity and mortality from emergency medical conditions. We surveyed providers at seven Nigerian Accident & Emergency (A&E) units about (i) their unit's ability to manage six major types of emergency medical condition (sentinel conditions) and (ii) barriers to performing key functions (signal functions) to manage sentinel conditions. Here, we present our analysis of provider-reported barriers to signal function performance.

### Methods

503 Health Providers at 7 A&E units, across 7 states, were surveyed using a modified African Federation of Emergency Medicine (AFEM) Emergency Care Assessment Tool (ECAT). Providers indicating suboptimal performance ascribed this performance to any of eight multiple-choice barriers [infrastructural issues, absent and broken equipment, inadequate training, inadequate personnel, requirement of out-of-pocket payment, non-indication of that signal function for the sentinel condition, and hospital-specific policies barring signal function performance] or an open-ended "other" response. The average number of endorsements for each barrier was obtained for each sentinel condition. Differences in barrier endorsement were compared across site, barrier type and sentinel condition using a three-way ANOVA test. Open-ended responses were evaluated using inductive thematic analysis. Sentinel conditions were Shock, Respiratory Failure, Altered Mental Status, Pain, Trauma, and Maternal & Child Health. Study sites were the University of Calabar Teaching Hospital, the Lagos University Teaching Hospital, the Federal Medical Center, Katsina, the National Hospital Abuja, the Federal Teaching Hospital Gombe, the University of Ilorin Teaching Hospital (Kwara), and the Federal Medical Center Owerri (Imo).

### Findings

Barrier distribution varied widely by study site. Just three study sites shared any one barrier to signal function performance as their most common. The two barriers most commonly

**Funding:** This work was funded by the Yale School of Medicine's Office of Student Research. The funder had no role in the study design, data collection, analysis, interpretation or writing of the report.

**Competing interests:** The authors have declared that no competing interests exist.

endorsed were (i) non-indication of, and (ii) insufficient infrastructure to perform signal functions. A three-way ANOVA test found significant differences in barrier endorsement by barrier type, study site and sentinel condition ($p < 0.05$). Thematic analysis of open-ended responses highlighted (i) considerations disfavoring signal function performance and (ii) lack of experience with signal functions as barriers to signal function performance. Interrater reliability, calculated using Fleiss' Kappa, was found to be 0.5 across 11 initial codes and 0.51 for our two final themes.

## Interpretation

Provider perspective varied with regards to barriers to care. Despite these differences, the trends seen for infrastructure reflect the importance of sustained investment in Nigerian health infrastructure. The high level of endorsement seen for the non-indication barrier may signal need for better ECAT adaptation for local practice & education, and for improved Nigerian emergency medical education and training. A low endorsement was seen for patient-facing costs, despite the high burden of Nigerian private expenditure on healthcare, indicating limited representation of patient-facing barriers. Analysis of open-ended responses was limited by the brevity and ambiguity of these responses on the ECAT. Further investigation is needed towards better representation of patient-facing barriers and qualitative approaches to evaluating Nigerian emergency care provision.

## Introduction

Emergency care is a crucial component of any healthcare system, and the development of high-quality and accessible emergency care in all countries has been identified as a key prerogative by organizations such as the WHO [1–4]. Access to emergency care in resource variable settings however, like access to other types of medical care, remains limited despite disproportionately high burdens of emergency conditions in these contexts [5–10].

An increasing number of countries in Africa have recognized this need and made strides towards developing emergency care systems, including the establishment of Emergency Medicine (EM) specialist training programs [11–15]. Nigeria, Africa's most populous country, is among those that have not yet done so, despite accounting for a large share of the continent's deaths and disability due to emergency medical conditions [16, 17]. Deaths due to Road Traffic Accidents in Nigeria occur at a rate of 1,042 deaths per 100,000 vehicles–one of the highest in the world [18]. The country is also experiencing several crises of violence that have led to the deaths of over 10,000 people in 2021 alone [19].

This data suggests a need for a thorough evaluation of Nigeria's emergency care system, to enable a more efficient allocation of resources to address acute and emergent injuries and conditions in the country. Currently there are limited data assessing the functionality of Nigerian A&E units, and the factors that contribute to deficits in their capacity for care. This study uses a cross-sectional analysis of provider survey responses to evaluate factors impeding care in Nigerian A&E units.

## Methods

### Study design and sites

For this cross-sectional study, we used www.random.org to randomly select six of twenty tertiary hospitals in Nigeria, representing each of Nigeria's six geopolitical zones. These were the

University of Calabar Teaching Hospital, the Lagos University Teaching Hospital, the Federal Medical Center, Katsina, the Federal Teaching Hospital Gombe, the University of Ilorin Teaching Hospital (Kwara), and the Federal Medical Center Owerri (Imo). In addition to these, a seventh facility at the National Hospital in the Federal Capital Territory, Abuja was purposively included for this center's dedicated trauma facility.

## Inclusion criteria

Survey data was collected using universal and convenience sampling. We took a universal approach to physicians, interviewing all doctors identified using rotation schedules as frequent A&E staff, and who were identified by the head of each hospital's A&E department as having worked in the A&E department within the three months preceding data collection. These doctors included resident doctors, medical officers (doctors yet to do their residency) and consultants (doctors who had completed their residency). Due to limitations in available resources to fund additional data collection, but a desire to enrich the perspectives represented in this study, we also interviewed a convenience sample of nurses who were non-rotating staff at each A&E, and who were sought out by data collectors at the time of each site visit. In African contexts, nurses may be critically important to the provision of emergency care, with crucial roles in the identification and triage of emergency medical conditions and patients, as well as direct patient care, making their perspectives particularly valuable for this type of analysis [7, 20]. Verbal consent was obtained from all participants.

## Tools

Our assesment uses the Emergency Care Assessment Tool (ECAT), designed by the African Federation of Emergency Medicine (AFEM) to evaluate the capacity of African emergency care facilities, and which has been used for this purpose in a number of African contexts [21]. The ECAT evaluates emergency medical care by assessing emergency units' provision of key medical interventions (signal functions) needed to treat six common emergency conditions (sentinel conditions). Groups of related signal functions correspond to specific sentinel conditions. The sentinel conditions assessed by the tool are Trauma (TRAUM), Respiratory failure (RESP), Altered Mental Status (AMS), Shock, Severe Pain (PAIN), and Maternal & Child Emergencies (MCH). A total of 73 signal functions were evaluated across all 6 conditions. Table 1 shows sentinel conditions and examples of their associated signal functions.

Our implementation of the ECAT was modified to match the scoring system used in the WHO Hospital Emergency Unit Assessment Tool (HEAT) [22]. The ECAT uses Yes/No

**Table 1. Sentinel conditions, samples of their associated sample signal functions, and the number of signal functions per sentinel condition.**

| Sentinel Condition | Sample Signal Functions | | Number of Signal Functions |
|---|---|---|---|
| **Respiratory Failure (RESP)** | Are manual maneuvers (e.g., jaw thrust, chin lift) done for obstructed airway in the emergency unit? | Are surgical airways created in the emergency unit? | 15 |
| **Maternal and Child Health (MCH)** | Are uterotonic drugs (e.g IV oxytocin) given in the emergency unit? | Are assisted vaginal deliveries (e.g with vacuum or forceps) done in the emergency unit? | 4 |
| **Shock (SHOCK)** | Are pelvic binders placed to control bleeding in the emergency unit? | Are central venous lines placed in the emergency unit? | 18 |
| **Altered Mental Status (AMS)** | Is blood glucose check and /or administered to hypoglycemic patients in the emergency unit? | Are lumbar punctures done in the emergency unit? | 10 |
| **Pain (PAIN)** | Are diagnostic and/or therapeutic paracenteses performed in the emergency unit? | Is Aspirin administered for acute coronary syndrome in the emergency unit? | 9 |
| **Trauma (TRAUM)** | Are chest tubes placed in the emergency unit? | Are thoracotomies done in the emergency unit? | 17 |

responses to measure signal function performance, with 'Yes' indicating signal function performance 'at least 90% of the time' and 'No' indicating performance 'less than 90% of the time' [23]. In our implementation, as in the WHO HEAT, we further stratified this system, reporting each signal function as "Generally Not Done", "Sometimes Done" and "Always Done". We also included questions reflecting context-specific aspects of Nigerian emergency care, such as respondent demographics and employment information. Of note, the WHO HEAT is not publicly available, but is accessible upon request to the WHO [22]. Our modified ECAT is included in S1 Appendix.

Reasons for non-optimal performance were elicited when performance for a signal function was reported as anything besides "Always Done". These were recorded using a multiple choice item with response options including (i) infrastructural issues, including such factors as unreliable electricity and lack of appropriate physical space to perform medical interventions, (ii) absent equipment, (iii) broken equipment, (iv) inadequate training, (v) inadequate personnel, (vi) requirement of out-of-pocket payment, (vii) non-indication of that signal function for the sentinel condition, (viii) hospital-specific policies barring signal function performance and (ix) an open-ended 'other' option that allowed participants to report barriers not otherwise included among these options. Respondents were permitted to endorse multiple barriers for each signal function.

Our analyses of signal function performance by site and sentinel condition are presented elsewhere. In this manuscript we assess barriers to performing these functions, including analyzing qualitative responses from open-ended components of the survey.

## Outcomes

Barriers to signal function performance were tallied across each sentinel condition. These tallies were then divided by the number of signal functions per sentinel condition to obtain averages reflecting the frequency with which these barriers were perceived as limiting adequate treatment for each sentinel condition at each study site and across the entire dataset.

## Data collection and analysis

Our modified ECAT was hosted on KoboToolbox, a tool developed by the Harvard Humanitarian Initiative for cloud-based data collection and storage in settings with limited internet access [24]. All data collection at each site was performed in-person by teams of two to three local medical students who received training on informed consent and interviewing techniques. Interviewers also had regular meetings with KU during the data collection period to discuss any concerns that emerged during data collection.

Survey data were analyzed using the R statistical software package to present the average number of times each barrier was endorsed as a reason for signal function failure by sentinel condition [25]. This was obtained by dividing the total number of responses for each barrier within each sentinel condition by the number of signal functions associated with each sentinel condition. A three-way ANOVA test was used to compare differences in the distributions of reasons provided for signal function failure across sentinel condition, study site and barrier type.

Open-ended elaborations provided for the "other" answer choice were treated with inductive thematic analysis. Interviews were cross-coded on Microsoft Excel using an adjusted structured tabular approach for thematic analysis of brief texts [26]. An initial codebook was developed by MM and iteratively expanded with cross-coding by CK, BF and AA such that each response was seen by at least two reviewers. Inductive thematic analysis based on final codes was then performed. Final themes were those that at least two reviewers agreed were

endorsed in at least five percent of open-ended responses. Interrater reliability was evaluated using the R statistical software package and Fleiss' Kappa. Writing and editing for this manuscript were performed by MM, KOO and CN.

### Ethical approval and role of funding source

This study was approved by the Human Research Protection Program Institutional Review Board, the National Health Research Ethics Committee of Nigeria, and the Ethical Review boards of each of the seven tertiary hospitals involved in the study. The funder, Yale School of Medicine's Office of Student Research, had no role in the study design, data collection, analysis, interpretation or writing of the report.

## Results

### Descriptive statistics

A total of 503 health care providers were interviewed across all 7 facilities, of which 390 were doctors, 110 were nurses and 3 did not indicate the specific type of health care worker they were. Bivariate analysis showed no significant difference in barrier endorsement between doctors and nurses (p = 0.643). Table 2 shows survey respondents by site and provider type.

Across all sites and sentinel conditions, an average of 77.5% of participants reported signal functions as "Always Done", reflecting optimal performance, while 21.1% of participants reported these as "Sometimes Done" or "Generally Not Done", reflecting suboptimal performance. Participant reports regarding signal function performance across all sites are shown by sentinel condition in Table 3. These results are discussed in more detail elsewhere.

Respondents provided 10,662 endorsements of barriers preventing signal function performance. The two most commonly endorsed barriers were non-indication of signal functions for sentinel conditions (23.6% of endorsed barriers), and insufficient infrastructure to perform signal functions (22.9% of endorsed barriers). These were followed by absent equipment (17.5%), inadequate personnel (14.2%), inadequate training (7.5%), "other" barriers not

**Table 2. Survey respondents by site and provider type.** Respondents listed as "Other" include consultants and those that did not provide a specific clinical role.

| Facility [State] | Providers | Provider Type | | | | | | |
|---|---|---|---|---|---|---|---|---|
| | | Registrar (Residents) | Nurse | Medical Officer (working full-time in the A&E) | Medical Officer (working part-time in the A&E) | Senior Registrar (Senior resident) | National Youth Corper | Other |
| **Federal Medical Center, Katsina [Katsina]** | 102 | 20 | 49 | 24 | 6 | 1 | 0 | 2 |
| **Lagos University Teaching Hospital [Lagos]** | 78 | 63 | 0 | 14 | 1 | 0 | 0 | 0 |
| **University of Calabar teaching Hospital [Cross River]** | 76 | 31 | 13 | 6 | 7 | 0 | 14 | 5 |
| **Federal Teaching Hospital Gombe [Gombe]** | 70 | 18 | 9 | 6 | 19 | 17 | 0 | 1 |
| **University of Ilorin Teaching Hospital [Kwara]** | 64 | 11 | 11 | 21 | 12 | 6 | 0 | 3 |
| **Federal Medical Center Owerri [Imo]** | 63 | 32 | 12 | 4 | 1 | 11 | 0 | 3 |
| **National Hospital Abuja [Abuja, Federal Capital Territory]** | 50 | 21 | 16 | 1 | 4 | 4 | 2 | 2 |
| **Total** | 503 | 196 | 110 | 76 | 50 | 39 | 16 | 16 |

**Table 3. Signal function performance by sentinel condition across all sites.** "Optimal" performance reflects that signal functions were reported as "Always Done", while "Suboptimal" performance reflects that signal functions were reported as "Sometimes Done" or "Generally Not Done". (MCH = Maternal and Child Health, RESP = Respiratory Failure, AMS = Altered Mental Status).

| Sentinel Condition | Performance | | | | | |
|---|---|---|---|---|---|---|
| | Suboptimal (%) | | Optimal (%) | | NA (%) | |
| MCH | 117 | (23.3) | 369 | (73.4) | 17 | (3.4) |
| TRAUMA | 115 | (22.9) | 377 | (75.0) | 11 | (2.2) |
| RESP | 157 | (31.2) | 342 | (68.0) | 4 | (0.8) |
| AMS | 66 | (13.1) | 434 | (86.3) | 3 | (0.6) |
| SHOCK | 96 | (19.1) | 405 | (80.5) | 2 | (0.4) |
| PAIN | 86 | (17.1) | 413 | (82.1) | 4 | (0.8) |
| All Sentinel Conditions | 637 | (21.1) | 2340 | (77.5) | 41 | (1.4) |

included among given multiple-choice responses (6.3%), broken equipment (5.1%), requirement of out-of-pocket patient payment (2.1%), and hospital policies barring signal function performance (0.8%).

The distribution of barrier endorsements varied widely by study site, with just three study sites [Lagos, Imo and Kwara] sharing any one barrier to signal function performance, insufficient infrastructure, as their most common. Additionally, while insufficient infrastructure was identified as the most frequently reported barrier to signal function performance at these sites, representing 43.7% of reported barriers among them, it represented just 3.4% of reported barriers at the study site in Abuja. A three-way ANOVA test found that differences in barrier endorsement by barrier type, study site and sentinel condition were significant ($p<0.05$). Barrier distributions across study sites and signal functions are shown in Table 4.

## Open-ended responses

Across all sites and sentinel conditions, "other" barriers not included within multiple choice options were endorsed 669 times (6.3% of all endorsed barriers). Of these endorsements, 541 included descriptions of these barriers that ranged from one to thirty words in length. Interrater reliability, calculated using Fleiss' Kappa, was calculated as 0.5 for all codes and 0.51 for final themes, indicating "moderate" agreement between reviewers [27].

Of 11 codes included in our initial codebook, two themes emerged that were endorsed in over 5% of responses by at least two reviewers. These were (i) non-indication of, or considerations disfavoring signal function performance (365 responses) and (ii) lack of experience seeing signal functions performed (58 responses), together covering 78.2% of "other" responses. The non-indication theme was further characterized by four common subthemes: (i) signal function obsolescence, (ii) hygiene and infection risk, (iii) outside referral and (iv) placement of the intervention outside the scope of the A&E unit. Five codes in our initial codebook were discarded as they did not align with themes meeting our 5% frequency threshold. These were: (i) bureaucratic barriers, (ii) inadequate equipment, (iii) a lack of patients presenting with sentinel conditions, (iv) personnel issues and (v) inadequate supplies.

Responses within the non-indication theme sometimes described signal functions recommended by the ECAT as "obsolete", describing other interventions that A&E providers should perform instead. Some providers, for example, described torniquet use for blood loss as typically deferred in favor of suture placement. Other signal functions, particularly those relating to rescue breathing, were described as not being performed as they were seen as "unhygienic" and carrying risk of infection, with some participants making note of COVID-19 as a specific infectious disease risk barring signal function performance. Other responses within this theme

**Table 4. Panels A through G show the average number of times each barrier was endorsed as a reason for signal function failure across sentinel conditions at (A) All Sites, (B) The University of Calabar Teaching Hospital [Cross River], (C) The Federal Medical Center, Katsina, (D) The Federal Teaching Hospital, Gombe, (E) The University of Ilorin Teaching Hospital [Kwara], (F) The Federal Medical Center, Owerri [Imo], (G) The National Hospital Abuja.**

| A—All Sites (503 Respondents) | | | | | | |
|---|---|---|---|---|---|---|
| Barrier Type | RESP | SHOCK | AMS | PAIN | TRAUM | MCH | Share of all endorsed barriers across all sentinel conditions (%) |
| Absent Equipment | 62.7 | 45.8 | 28.9 | 23.2 | 38.2 | 8.0 | 17.5 |
| Fees | 3.3 | 4.7 | 1.1 | 0.9 | 13.9 | 0.6 | 2.1 |
| Non-Indication | 85.1 | 82.7 | 13.9 | 29.2 | 53.2 | 15.8 | 23.6 |
| Infrastructure | 70.6 | 46.0 | 26.9 | 26.3 | 76.4 | 25.3 | 22.9 |
| Hospital Policy | 8.1 | 0.2 | 0.2 | 0.0 | 0.8 | 0.0 | 0.8 |
| Other | 36.2 | 10.2 | 4.9 | 6.3 | 12.1 | 4.6 | 6.3 |
| Limited Personnel | 49.4 | 27.2 | 5.2 | 1.9 | 80.0 | 4.8 | 14.2 |
| Broken Equipment | 18.8 | 19.7 | 4.8 | 9.1 | 6.2 | 1.9 | 5.1 |
| Limited Training | 24.1 | 23.2 | 2.2 | 1.2 | 36.7 | 1.9 | 7.5 |

| B—Cross River (76 Respondents) | | | | | | |
|---|---|---|---|---|---|---|
| Barrier Type | RESP | SHOCK | AMS | PAIN | TRAUM | MCH | Share of all endorsed barriers across all sentinel conditions (%) |
| Absent Equipment | 18.4 | 12.6 | 11.4 | 11.2 | 4.3 | 0.1 | 15.6 |
| Fees | 2.4 | 2.4 | 0.1 | 0.8 | 8.1 | 0.1 | 3.7 |
| Non-Indication | 35.1 | 38.3 | 2.4 | 15.0 | 24.6 | 3.1 | 31.8 |
| Infrastructure | 27.3 | 20.7 | 11.8 | 9.8 | 15.8 | 0.0 | 22.9 |
| Hospital Policy | 0.8 | 0.0 | 0.0 | 0.0 | 0.0 | 0.0 | 0.2 |
| Other | 4.8 | 0.4 | 0.0 | 0.0 | 0.2 | 0.0 | 1.5 |
| Limited Personnel | 18.3 | 14.4 | 4.3 | 0.8 | 21.6 | 0.3 | 16.0 |
| Broken Equipment | 3.1 | 1.3 | 0.2 | 5.7 | 0.6 | 0.0 | 2.9 |
| Limited Training | 4.8 | 4.0 | 0.9 | 0.2 | 10.1 | 0.2 | 5.4 |

| C–Katsina (102 Respondents) | | | | | | |
|---|---|---|---|---|---|---|
| Barrier Type | RESP | SHOCK | AMS | PAIN | TRAUM | MCH | Share of all endorsed barriers across all sentinel conditions (%) |
| Absent Equipment | 2.3 | 14.3 | 0.4 | 0.1 | 10.2 | 0.0 | 23.4 |
| Fees | 0.0 | 0.1 | 0.4 | 0.0 | 2.6 | 0.0 | 2.7 |
| Non-Indication | 8.1 | 1.9 | 1.0 | 0.1 | 2.3 | 0.2 | 11.7 |
| Infrastructure | 0.7 | 1.4 | 0.6 | 0.2 | 10.1 | 0.0 | 11.1 |
| Hospital Policy | 0.3 | 0.0 | 0.0 | 0.0 | 0.3 | 0.0 | 0.6 |
| Other | 6.2 | 1.0 | 0.6 | 0.7 | 1.9 | 0.1 | 8.9 |
| Limited Personnel | 0.8 | 0.8 | 0.3 | 0.0 | 9.9 | 0.0 | 10.1 |
| Broken Equipment | 2.0 | 13.6 | 0.3 | 0.1 | 3.9 | 0.0 | 17.0 |
| Limited Training | 0.8 | 7.0 | 0.4 | 0.0 | 8.8 | 0.0 | 14.5 |

| D–Gombe (70 Respondents) | | | | | | |
|---|---|---|---|---|---|---|
| Barrier Type | RESP | SHOCK | AMS | PAIN | TRAUM | MCH | Share of all endorsed barriers across all sentinel conditions (%) |
| Absent Equipment | 23.0 | 4.2 | 6.6 | 1.0 | 2.7 | 4.2 | 15.5 |
| Fees | 0.4 | 0.4 | 0.1 | 0.0 | 1.9 | 0.4 | 1.2 |
| Non-Indication | 9.8 | 14.9 | 0.9 | 7.1 | 7.0 | 11.9 | 19.1 |
| Infrastructure | 9.0 | 2.7 | 1.2 | 0.3 | 7.3 | 1.9 | 8.3 |
| Hospital Policy | 3.2 | 0.0 | 0.0 | 0.0 | 0.1 | 0.0 | 1.2 |
| Other | 1.2 | 0.0 | 0.1 | 0.4 | 0.6 | 0.7 | 1.1 |
| Limited Personnel | 27.1 | 9.0 | 0.3 | 0.2 | 46.6 | 4.3 | 32.5 |
| Broken Equipment | 12.2 | 1.2 | 1.8 | 0.0 | 0.8 | 1.1 | 6.3 |
| Limited Training | 15.1 | 7.3 | 0.3 | 0.2 | 15.0 | 1.7 | 14.7 |

| E–Lagos (78 Respondents) | | | | | | |
|---|---|---|---|---|---|---|
| Barrier Type | RESP | SHOCK | AMS | PAIN | TRAUM | MCH | ALL | Share of all endorsed barriers across all sentinel conditions (%) |
| Absent Equipment | 2.7 | 1.2 | 0.6 | 0.2 | 10.9 | 3.7 | 19.2 | 20.6 |
| Fees | 0.0 | 0.1 | 0.2 | 0.0 | 0.7 | 0.0 | 1.0 | 1.1 |

*(Continued)*

**Table 4.** (Continued)

| Non-Indication | 0.9 | 1.0 | 0.7 | 0.6 | 0.9 | 0.1 | 4.1 | 4.4 |
|---|---|---|---|---|---|---|---|---|
| Infrastructure | 3.6 | 2.7 | 0.4 | 0.7 | 26.8 | 13.7 | 47.8 | 51.3 |
| Hospital Policy | 0.1 | 0.0 | 0.0 | 0.0 | 0.1 | 0.0 | 0.2 | 0.2 |
| Other | 7.6 | 1.0 | 0.9 | 0.7 | 4.9 | 1.1 | 16.1 | 17.3 |
| Limited Personnel | 0.6 | 0.0 | 0.0 | 0.2 | 0.4 | 0.1 | 1.3 | 1.4 |
| Broken Equipment | 0.2 | 0.0 | 0.0 | 0.0 | 0.6 | 0.7 | 1.5 | 1.6 |
| Limited Training | 0.7 | 0.7 | 0.1 | 0.2 | 0.2 | 0.0 | 1.9 | 2.0 |

| F–Kwara (64 Respondents) | | | | | | | |
|---|---|---|---|---|---|---|---|
| **Barrier Type** | **RESP** | **SHOCK** | **AMS** | **PAIN** | **TRAUM** | **MCH** | **Share of all endorsed barriers across all sentinel conditions (%)** |
| Absent Equipment | 11.4 | 8.4 | 6.7 | 9.4 | 6.8 | 0.0 | 23.0 |
| Fees | 0.3 | 1.4 | 0.1 | 0.0 | 0.6 | 0.0 | 1.3 |
| Non-Indication | 16.4 | 13.4 | 4.8 | 2.9 | 8.3 | 0.0 | 24.7 |
| Infrastructure | 22.1 | 12.4 | 7.2 | 8.7 | 10.3 | 0.0 | 32.7 |
| Hospital Policy | 3.4 | 0.0 | 0.2 | 0.0 | 0.0 | 0.0 | 2.0 |
| Other | 6.8 | 3.0 | 0.7 | 1.2 | 1.3 | 0.1 | 7.1 |
| Limited Personnel | 1.7 | 1.2 | 0.1 | 0.7 | 1.2 | 0.0 | 2.6 |
| Broken Equipment | 0.4 | 2.8 | 1.8 | 3.1 | 0.0 | 0.0 | 4.4 |
| Limited Training | 1.0 | 1.2 | 0.3 | 0.6 | 1.1 | 0.0 | 2.3 |

| G–Imo (63 Respondents) | | | | | | | |
|---|---|---|---|---|---|---|---|
| **Barrier Type** | **RESP** | **SHOCK** | **AMS** | **PAIN** | **TRAUM** | **MCH** | **Share of all endorsed barriers across all sentinel conditions (%)** |
| Absent Equipment | 2.8 | 1.9 | 2.9 | 1.1 | 2.7 | 0.0 | 13.3 |
| Fees | 0.0 | 0.0 | 0.0 | 0.0 | 0.0 | 0.0 | 0.0 |
| Non-Indication | 3.7 | 2.8 | 0.8 | 0.3 | 3.7 | 0.0 | 13.2 |
| Infrastructure | 7.8 | 5.1 | 5.6 | 6.7 | 5.2 | 9.8 | 47.1 |
| Hospital Policy | 0.1 | 0.1 | 0.0 | 0.0 | 0.0 | 0.0 | 0.3 |
| Other | 8.0 | 3.1 | 1.6 | 2.0 | 2.3 | 0.0 | 20.0 |
| Limited Personnel | 0.1 | 1.0 | 0.0 | 0.0 | 0.2 | 0.0 | 1.6 |
| Broken Equipment | 0.3 | 0.2 | 0.7 | 0.2 | 0.1 | 0.1 | 2.0 |
| Limited Training | 0.3 | 1.2 | 0.0 | 0.0 | 0.7 | 0.0 | 2.6 |

| H–Abuja (50 Respondents) | | | | | | | |
|---|---|---|---|---|---|---|---|
| **Barrier Type** | **RESP** | **SHOCK** | **AMS** | **PAIN** | **TRAUM** | **MCH** | **Share of all endorsed barriers across all sentinel conditions (%)** |
| Absent Equipment | 2.0 | 3.1 | 0.3 | 0.1 | 0.7 | 0.0 | 10.2 |
| Fees | 0.1 | 0.1 | 0.1 | 0.1 | 0.1 | 0.0 | 0.9 |
| Non-Indication | 11.1 | 10.3 | 3.3 | 3.2 | 6.4 | 0.4 | 57.4 |
| Infrastructure | 0.1 | 1.0 | 0.1 | 0.0 | 0.9 | 0.0 | 3.5 |
| Hospital Policy | 0.1 | 0.1 | 0.0 | 0.0 | 0.2 | 0.0 | 0.7 |
| Other | 1.7 | 1.7 | 1.1 | 1.3 | 0.9 | 2.6 | 15.2 |
| Limited Personnel | 0.9 | 0.8 | 0.1 | 0.0 | 0.1 | 0.0 | 3.1 |
| Broken Equipment | 0.4 | 0.6 | 0.0 | 0.0 | 0.3 | 0.0 | 2.2 |
| Limited Training | 1.4 | 1.8 | 0.1 | 0.0 | 0.8 | 0.0 | 6.8 |

placed certain signal functions outside the scope of care at A&E units. Surgical airway creation, for example, was described as requiring referral to surgery, anesthesia, or the ICU, and many signal functions related to Maternal & Child health were described as typically referred to OB-GYN for management. Other procedures and interventions, such as specific lab and imaging studies, were additionally described as typically outsourced to outside facilities. Of all responses endorsing signal function non-indication, 261 (71%) were given to signal functions relating to the Respiratory Failure sentinel condition.

**Table 5. Final themes, selected responses endorsing these themes, and corresponding subthemes.** Participant responses ranged from one to thirty words in length, and 87% of responses were under ten words in length.

| Theme | Survey Question | Sample Participant Responses | Respondent Role | Subthemes |
|---|---|---|---|---|
| Non-Indication & Considerations against Signal Function Performance | Is there access to emergency surgical services (e.g. caesarean section) in the emergency unit? | *Generally done at obstetric theater* | Senior Registrar | Beyond the scope of the A&E unit |
| | Are surgical airways created in the emergency unit? | *The Anesthetics or ENT surgeons have to be invited for such kind of procedures* | Medical Officer (Working Part-Time in the A&E) | |
| | Are Endotracheal intubations done in the emergency unit? | *This procedure is done in the ICU* | Registrar | |
| | Are tourniquets placed to control bleeding in the emergency unit? | *Can lead to complications so it has to be timed to avoid ischemia. Sutures done in place* | Senior Registrar | Other interventions preferred |
| | In the setting of a pneumothorax, is a three-way dressing done in the emergency unit? | *Thoracentesis is preferred* | Medical Officer (Working Full-Time in the A&E) | |
| | Is Rescue Breathing (Mouth-to-Mouth Resuscitation) done in the emergency unit? | *We use Ambu Bag, so no need for mouth-mouth breathing* | Registrar | |
| | | *Not hygienic* | Medical Officer (Working Part-Time in the A&E) | Hygiene & Infection Risk |
| | | *Risk of COVID and other infections* | Registrar | |
| | Is Urine Dipstick (urinalysis) performed in the emergency unit? | *Patients are usually referred to the lab for the test. The medical emergency unit seldom does it* | Registrar | Referred to outside facility |
| | Are Chest X-Rays done in the emergency unit? | *Patients are referred to radiology or labs in town* | Registrar | |
| | Is there access to neurosurgical services in the emergency or trauma unit at your hospital? | *Neurosurgeons are being recruited here, so far for now mostly we refer to Kano [hospital in another city]* | Medical Officer (Working Full-Time in the A&E) | |
| Signal Function not Witnessed | Are surgical airways created in the emergency unit? | *Haven't witnessed it* | Registrar | Unwitnessed |
| | In the setting of a pneumothorax, is a three-way dressing done in the emergency unit? | *No insight about it* | Medical Officer (Working Full-Time in the A&E) | |
| | Are Autotransfusions from chest tubes done in the emergency unit? | *I've never seen one done here* | Registrar | |

Final themes, subthemes, selected responses endorsing these, and the clinical roles of respondents providing these endorsements are shown in Table 5.

Notably, of the 58 responses endorsing a lack of experience seeing signal functions performed, forty were provided by registrars and senior registrars, both categories of doctors still in training. Additionally, in 31 "other" responses, respondents described that signal functions were performed in their A&E units, but only "when indicated".

## Discussion & limitations

Our study provides an assessment of provider endorsed barriers to the delivery of emergency medical care in Nigerian A&E units. To our knowledge, this is the first assessment of its kind, and the largest of any study around emergency medicine in Nigeria to date, and provides key guidance on improving capacity for emergency care in the country.

Despite considerable differences in reported barriers by site and sentinel condition, descriptive statistical analysis highlighted both non-indication and infrastructural deficits as key contributors to the non-performance of signal functions among our study sites. Open-ended

response analysis further identified non-indication and considerations guiding against signal functions, such as being outside of the scope of ED practice, as prominent barriers to signal function performance in this study.

The ECAT is a survey tool designed for use in African contexts. That non-indication remained one of the most commonly identified reasons for non-performance of signal functions despite this raises concern about the ECAT's compatibility with Nigerian medical education, training, and practice. Previous studies using the ECAT to assess the quality of emergency care have found the non-indication barrier endorsed considerably less prominently [23, 28–31]. The version of the ECAT deployed for this study was one that had undergone one series of refinement following the tool's initial release, adjusting for features of emergency care in Cameroon, Uganda, Egypt, Botswana and South Africa, to reflect factors affecting practice throughout the African continent [21]. It is possible that some aspects of this refinement still do not generalize to Nigeria, where healthcare has been shaped by a unique pre- and post-colonial history, and whose modern institutions may not be directly comparable to those of other African states [32, 33]. That our study was conducted at the height of the COVID-19 pandemic may complicate this picture further, giving providers greater pause regarding signal functions that might otherwise be performed in the absence of a global respiratory pandemic. This context may account for the high number of "non-indication" endorsements provided for signal functions corresponding to the respiratory failure sentinel condition. Other research has noted the wide-reaching impact of COVID-19 on emergency medical practice in a variety of settings [34–38]. This suggests a need for more thorough adaptation of the ECAT for local contexts, taking into consideration not only stable characteristics of care in these contexts, but more dynamic factors which may affect medical practice as well.

Alternatively, the high level of endorsement seen for the non-indication barrier may demonstrate suboptimal emergency medicine education and training. Nigeria has only recently established an emergency medicine training program for physicians, and Nigerian A&E units are often staffed by physicians with little to no emergency training [39, 40]. Many such providers may be unexposed to various aspects of emergency medical practice, and thus more likely to erroneously view emergency medicine interventions as non-indicated. This demonstrates continued need for improved emergency medicine training in Nigeria. Recent WHO resolutions have supported this, urging that member states create and maintain dedicated emergency medicine specialist training programs [41, 42]. In Nigeria, this work may be supported by organizations such as the Nigeria Emergency Medicine Society, African Federation for Emergency Medicine, and International Federation for Emergency Medicine, which are local, regional, and international bodies, respectively, that can provide consultation and support best practices that have been implemented in similar settings [43, 44]. The WHO emergency care framework may additionally provide a roadmap for effective strategies to guide emergency care development in Nigeria [45]. In the interim, stopgap measures such as the WHO Basic Emergency Course, which has been piloted in Nigerian settings, may be considered as alternative means to increase provider proficiency in emergency medical practice [46].

That deficits in infrastructure were also frequently endorsed as barriers at our study sites brings attention to resource-related limitations on Nigerian emergency care. Factors reflecting physical and human resources––infrastructure, equipment, personnel, training, and supplies––made up 67.2% of endorsed barriers in this study. This reflects a problem of resource scarcity that permeates Nigerian healthcare and the Nigerian economy more broadly, and that has been exacerbated by the COVID-19 pandemic, which has seen the worst performance of the country's economy since its 1999 return to democracy [47–50]. A deteriorating exchange rate contributes to this, increasing the local cost of crucial foreign inputs into the Nigerian economy [51, 52]. These inputs include medical devices, of which Nigeria's supply is largely

imported [53]. All of these risk worsening supply-side pressures on the Nigerian health industry and worsening resource limitations in Nigerian emergency care, and highlight the need for concerted action against causes of economic instability and towards greater investment in the Nigerian health sector.

It is also worth noting that our results reflect the perspective of Nigerian emergency medicine providers, and may not reflect barriers to care experienced by people with other roles in A&E units. In particular, the low level of endorsement seen for patient facing costs comes at odds with data suggesting that Nigerian patients experience one of the highest rates of private expenditure on healthcare in West Africa, with out-of-pocket costs accounting for 70.5% of all Nigerian healthcare expenditure in 2019, compared to an average of 47.3% for West Africa [54]. Other research has demonstrated that this high burden of private expenditure serves as a barrier to health access for patients across a wide variety of conditions [55–61]. That patient fees made up just 2.1% of endorsed barriers in this study suggests the ECAT and its use in this study may not adequately reflect patient-facing barriers to care, which are of vital importance in improving access to emergency medical care [62]. Further investigation on the relationship between patient healthcare costs and use of emergency medical services in Nigeria are needed, with consideration given to strategies for mitigating these costs.

Finally, our open-ended response analysis of open-ended descriptions of "other" barriers not included within multiple choice options provided additional insight but was limited by response brevity, with 87% of responses being under ten words in length. This brevity caused some ambiguity in the interpretation of these responses. For example, respondents who reported that signal functions were not performed for "hygiene" may have been alluding to non-indication and considerations disfavoring the performance of signal functions, but may alternatively have meant that their facilities lack adequate equipment and supplies to guarantee that these functions are performed hygienically. This ambiguity may have limited interrater reliability for initial codes and final themes, which, in both cases, was found to only be moderate. This is further illustrated in the 31 open-ended responses describing that signal functions were performed 'when indicated'–suggesting appropriate performance–despite previously noting these signal functions as performed suboptimally. The brevity of these responses limits our ability to resolve this type of discordance. A more robust qualitative design, though not the primary approach used in this study, may be more able to contextualize provider perspectives around additional barriers to care, and provide more nuanced findings.

## Conclusion

This evaluation of provider-reported barriers to emergency care at Nigerian A&E units suggests that suboptimal performance on the ECAT may be driven by potential differences between ECAT signal functions and Nigerian medical practice and training, a lack of dedicated emergency medical education in Nigeria, and by resource-related deficits in the Nigerian health sector that derive from socioeconomic factors driving underperformance across many aspects of Nigerian society. It highlights a need for intervention at all of these intersections, and supports future work considering patient-facing concerns and using robust qualitative approaches to further investigate barriers preventing the optimal delivery of emergency care in Nigeria.

## Supporting information

**S1 Appendix. Assessment of emergency care services in Nigeria (AFEM Tool).**
(PDF)

## Author Contributions

**Conceptualization:** Muzzammil Imran Muhammad, Kelechi Umoga, Kehinde Olawale Ogunyemi, Christine Ngaruiya.

**Data curation:** Muzzammil Imran Muhammad, Kelechi Umoga.

**Formal analysis:** Muzzammil Imran Muhammad, Amber Acquaye, Brian Fleischer, Chigoziri Konkwo.

**Funding acquisition:** Kelechi Umoga.

**Investigation:** Muzzammil Imran Muhammad, Kelechi Umoga.

**Methodology:** Muzzammil Imran Muhammad.

**Project administration:** Muzzammil Imran Muhammad.

**Visualization:** Muzzammil Imran Muhammad.

**Writing – original draft:** Muzzammil Imran Muhammad.

**Writing – review & editing:** Muzzammil Imran Muhammad, Kelechi Umoga, Amber Acquaye, Brian Fleischer, Chigoziri Konkwo, Kehinde Olawale Ogunyemi, Christine Ngaruiya.

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
