## [Decision Letter · Decision Letter 0]

15 Mar 2023

PGPH-D-23-00106

Provider-Identified Barriers to Performance at Seven Nigerian Accident & Emergency Units: A Cross-Sectional Study

Dear Dr. Imran Muhammad,

Thank you for submitting your manuscript to PLOS Global Public Health. After careful consideration, we feel that it has merit but does not fully meet PLOS Global Public Health’s publication criteria as it currently stands. Therefore, we invite you to submit a revised version of the manuscript that addresses the points raised during the review process.

The reviewers' comments are appended below. I ask that you respond to each of these comments as fully as possible. Pay particular attention to comments by Reviewer 2.

We look forward to receiving your revised manuscript.

Kind regards,

Elize Massard da Fonseca, Ph.D.

Academic Editor

Journal Requirements:

Additional Editor Comments (if provided):

Reviewers' comments:

Reviewer's Responses to Questions

**Comments to the Author**

1. Does this manuscript meet PLOS Global Public Health’s publication criteria? Is the manuscript technically sound, and do the data support the conclusions? The manuscript must describe methodologically and ethically rigorous research with conclusions that are appropriately drawn based on the data presented.

Reviewer #1: Yes

Reviewer #2: Partly

2. Has the statistical analysis been performed appropriately and rigorously?

Reviewer #1: Yes

Reviewer #2: Yes

3. Have the authors made all data underlying the findings in their manuscript fully available (please refer to the Data Availability Statement at the start of the manuscript PDF file)?

Reviewer #1: Yes

Reviewer #2: Yes

4. Is the manuscript presented in an intelligible fashion and written in standard English?

Reviewer #1: Yes

Reviewer #2: Yes

5. Review Comments to the Author

Reviewer #1: This mixed methods paper investigated factors impeding provision of quality emergency care in health facilities in Nigeria. This is an important topic and very well addressed by the authors. Faced with limited number of trained personnel and high burden of accidents and violence, African countries need to improve on the quality of emergency services available for patients.

Reviewer #2: The identification of barriers to adequate healthcare access is always a relevant concern, and the accident and emergency care units play an important role in the health services’ network. On the other hand, the manuscript is clear, and the discussion presented by the authors based on the findings sounds pertinent and useful. However, some aspects need to be better calibrated in the paper:

1. In composing the sample of respondents with all doctors and a convenience sample of nurses, did the authors consider the implications for statistical representativeness. From whom did they want to capture the responses? Doctors and nurses in the seven healthcare units selected? In this case, wouldn't it be necessary to attribute sample weights different from one to nurses?

2. Weren’t there losses from the sample designed?

3. Wouldn’t it be possible to list the signal functions for the sentinel conditions in a Box?

4. In Results, it does not sound appropriate the sum of individuals in Table 2 for ALL. Consequently, it would be more adequate, in the second paragraph of the section to say that a mean of 77.5% of participants reported signal functions as “Always Done”, reflecting optimal performance, while a mean of 21.1% of participants reported these as “Sometimes Done” or “Generally Not Done”, reflecting suboptimal performance.

5. Table 3, broken in eight fragments which are, in fact, also tables themselves, does not seem OK for me. I would prefer to have a solely table.

6. Another point about Table 3 has to do with the lack of the number of respondents in each column (for all and each A&E unit), that, varies. It is a relevant information.

7. Reference 24 is cited as a complementary source of information about the work done, but it has not been published yet.

6. PLOS authors have the option to publish the peer review history of their article (what does this mean?). If published, this will include your full peer review and any attached files.

**Do you want your identity to be public for this peer review?** For information about this choice, including consent withdrawal, please see our Privacy Policy.

Reviewer #1: No

Reviewer #2: No

---

## [Decision Letter · Decision Letter 1]

20 Apr 2023

Provider-Identified Barriers to Performance at Seven Nigerian Accident & Emergency Units: A Cross-Sectional Study

PGPH-D-23-00106R1

Dear Muzzammil Imran Muhammad,

We are pleased to inform you that your manuscript 'Provider-Identified Barriers to Performance at Seven Nigerian Accident & Emergency Units: A Cross-Sectional Study' has been provisionally accepted for publication in PLOS Global Public Health.

Best regards,

Elize Massard da Fonseca, Ph.D.

Academic Editor

Reviewer Comments (if any, and for reference):

Reviewer's Responses to Questions

**Comments to the Author**

1. If the authors have adequately addressed your comments raised in a previous round of review and you feel that this manuscript is now acceptable for publication, you may indicate that here to bypass the “Comments to the Author” section, enter your conflict of interest statement in the “Confidential to Editor” section, and submit your "Accept" recommendation.

Reviewer #2: All comments have been addressed

2. Does this manuscript meet PLOS Global Public Health’s publication criteria? Is the manuscript technically sound, and do the data support the conclusions? The manuscript must describe methodologically and ethically rigorous research with conclusions that are appropriately drawn based on the data presented.

Reviewer #2: Yes

3. Has the statistical analysis been performed appropriately and rigorously?

Reviewer #2: Yes

4. Have the authors made all data underlying the findings in their manuscript fully available (please refer to the Data Availability Statement at the start of the manuscript PDF file)?

Reviewer #2: Yes

5. Is the manuscript presented in an intelligible fashion and written in standard English?

Reviewer #2: Yes

6. Review Comments to the Author

Reviewer #2: Although some of my comments regarding sampling were not truly understood, I believe that the authors addressed adequately the most critical points. I think that the manuscript attends publication criteria. A problem persists in telation to the format of the tables, which seem truncated in the file made available.

7. PLOS authors have the option to publish the peer review history of their article (what does this mean?). If published, this will include your full peer review and any attached files.

**Do you want your identity to be public for this peer review?** For information about this choice, including consent withdrawal, please see our Privacy Policy.

Reviewer #2: No
